



# Aphotic N₂ fixation along an oligotrophic to ultraoligotrophic transect in the Western Tropical South Pacific Ocean

Mar Benavides[1,2], Katyanne M. Shoemaker[3], Pia H. Moisander[3], Jutta Niggemann[4], Thorsten Dittmar[4],

Solange Duhamel[5], Olivier Grosso[6], Mireille Pujo-Pay[7], Sandra Hélias-Nunige[6], Sophie Bonnet[1]

[1]Aix Marseille Université, CNRS/INSU, Université de Toulon, IRD, Mediterranean Institute of Oceanography (MIO) UM 110, 98848 Nouméa, New Caledonia

[2]Marine Biology Section, Department of Biology, University of Copenhagen, 3000 Helsingør, Denmark

[3]Department of Biology, University of Massachusetts Dartmouth, 285 Old Westport Road, North Dartmouth, MA 02747, USA

[4]Research Group for Marine Geochemistry (MPI-ICBM Bridging Group), Institute for Chemistry and Biology of the Marine Environment University of Oldenburg, Carl-von-Ossietzky-Strasse 9-11, D-26129 Oldenburg, Germany

[5]Lamont-Doherty Earth Observatory, Division of Biology and Paleo Environment, Columbia University, PO Box 1000, 61 Route 9W, Palisades, NJ 10964, USA

[6]Aix Marseille Université, CNRS/INSU, Université de Toulon, IRD, Mediterranean Institute of Oceanography (MIO) UM 110, 13288 Marseille, France

[7]Laboratoire d'Océanographie Microbienne – UMR 7321, CNRS - Sorbonne Universités, UPMC Univ

Paris 06, Observatoire Océanologique, 66650 Banyuls-sur-mer, France

*Correspondence to*: Mar Benavides (mar.benavides@bio.ku.dk)

**Abstract.**

The western tropical South Pacific (WTSP) Ocean has been recognized as a global hotspot of dinitrogen (N₂) fixation. Here, as in other marine environments across the oceans, N₂ fixation studies have focused in the sunlit layer. However, studies have confirmed the importance of aphotic N₂ fixation activity, although until now only one had been performed in the WTSP. In order to increase our knowledge of aphotic N₂ fixation in the WTSP, here we measure N₂ fixation rates and identify diazotrophic

phylotypes in the mesopelagic layer along a transect spanning from New Caledonia to French Polynesia. Because non-cyanobacterial diazotrophs presumably need external dissolved organic matter (DOM) sources for their nutrition, we also identified DOM compounds using Fourier Transform Ion Cyclotron Mass Spectrometry (FTICRMS). N₂ fixation rates were low (average $0.63 \pm 0.07$ nmol N L⁻¹ d⁻¹), but consistently detected across all depths and stations, representing ~6-88% of photic N₂ fixation.

N₂ fixation rates were not significantly correlated to DOM compounds. The analysis of *nifH* gene




amplicons revealed a wide diversity of non-cyanobacterial diazotrophs, majorly matching clusters 1 and 3. Interestingly, a distinct phylotype from the major *nifH* subcluster 1G dominated at 650 dbar, coinciding with the oxygenated Sub-Antarctic Mode Water (SAMW). This consistent pattern suggests that the distribution of aphotic diazotroph communities is to some extent controlled by water mass

structure. While the data available is still too scarce to elucidate the distribution and controls of mesopelagic non-cyanobacterial diazotrophs in the WTSP, their prevalence in the mesopelagic layer and the consistent detection of active $N_2$ fixation activity at all depths sampled during our study suggest that aphotic $N_2$ fixation may contribute significantly to fixed nitrogen inputs in this area.

**1 Introduction**

Pelagic $N_2$ fixation is considered the greatest input of fixed nitrogen to the oceans, adding up to ~100-107 Tg N per year (Codispoti, 2007; Galloway et al., 2004; Gruber and Galloway, 2008; Jickells et al., 2017). In the sunlit layer of the warm oligotrophic tropical and subtropical oceans, cyanobacterial diazotrophs such as *Trichodesmium*, UCYN-B and diatom-diazotroph associations (DDAs) dominate

fixed nitrogen inputs via $N_2$ fixation (Capone et al., 1997; Montoya et al., 2004). In colder and less oligotrophic waters at higher latitudes, other diazotrophs including UCYN-A and non-cyanobacterial groups such as Gamma A may be more competitive (Bonnet et al., 2015; Langlois et al., 2015; Moisander et al., 2010; 2014), expanding considerably the latitudinal range where $N_2$ fixation is considered significant in predictive biogeochemical models. In the past decade, several studies have

retrieved *nifH* sequences from the dark ocean, some also accompanied by low $N_2$ fixation rates (Bonnet et al., 2013; Hamersley et al., 2011; Rahav et al., 2013). Due to the immense volume of the dark ocean, aphotic $N_2$ fixation could influence the global nitrogen budget substantially. However, the number of published aphotic $N_2$ fixation rates is scant and our understanding of aphotic diazotrophs' metabolism and ecology of aphotic diazotrophs is still limited, hindering our ability to evaluate their impact on

global fixed nitrogen inputs (Moisander et al., 2017).

Non-cyanobacterial diazotrophs span the four established *nifH* gene clusters (Chien and Zinder, 1996), are the most numerous in *nifH* gene databases (Farnelid and Riemann, 2008), and are spread throughout the global ocean (Bonnet et al., 2015; Farnelid et al., 2011; Langlois et al., 2015; Messer et al., 2015). As discussed in Bombar et al. (2016), the growth and activity of non-cyanobacterial

diazotrophs may be controlled by (i) the presence of oxygen -because oxygen destroys the nitrogenase enzyme, (ii) the availability of fixed nitrogen because $N_2$ fixation becomes too energetically expensive when reduced nitrogen forms are readily available in the environment, and (iii) the availability of energy because non-cyanobacterial diazotrophs may not be able to photosynthesize and thus rely on external fixed carbon sources. However, aphotic diazotrophic activity has been found both in oxygen

deficient regions such as the oxygen minimum zone of the Eastern Tropical South Pacific (Bonnet et al., 2013; Loescher et al., 2014), and fully oxygenated mesopelagic waters in the Mediterranean Sea



(Benavides et al., 2016; Rahav et al., 2013). Moreover, while fixed nitrogen availability should theoretically shut down $N_2$ fixation, significant $N_2$ fixation rates have been measured in nitrate rich mesopelagic waters of the Western Tropical South Pacific (WTSP) (Benavides et al., 2015). Finally, energy is likely made available to heterotrophic non-cyanobacterial diazotrophs through labile dissolved

organic matter (DOM). Aphotic $N_2$ fixation rates have been related to the presence of relatively labile DOM compounds such as transparent exopolymeric particles (TEP) in the WTSP (Benavides et al., 2015), or peptides and unsaturated aliphatics in the Mediterranean Sea (Benavides et al., 2016). The addition of small labile DOM molecules such as carbohydrates or amino acids has been shown to enhance aphotic $N_2$ fixation in various environments (Benavides et al., 2015; Bonnet et al., 2013;

Loescher et al., 2014; Rahav et al., 2013). However, some photic non-cyanobacterial diazotrophs also bear genes for the degradation of refractory DOM compounds (e.g. aromatic hydrocarbons; Bentzon-Tilia et al., 2015). It is thus reasonable to expect that aphotic non-cyanobacterial diazotrophs may be able to exploit diverse DOM sources. Unfortunately, the current lack of genome information from non-cyanobacterial aphotic diazotrophs does not allow us to assess how they are affected by DOM

composition and lability.

        The WTSP has been recently recognized as a global hotspot of photic $N_2$ fixation, harboring among the highest $N_2$ fixation rates ever recorded (~600 $\mu$mol N m$^{-2}$ d$^{-1}$; Bonnet et al., 2017), mostly attributed to *Trichodesmium* and UCYN-B (Berthelot et al., 2017; Bonnet et al., 2015; 2009; Stenegren et al., 2017). To the eastern border of this region, the ultraoligotrophic South Pacific Gyre (GY) has low

photic $N_2$ fixation rates (Raimbault and Garcia, 2008), which have been mainly attributed to small unicellular diazotrophs such as UCYN-A and Gammaproteobacteria (Bonnet et al., 2008; Halm et al., 2012; Stenegren et al., 2017). Despite its potentially immense implications in global fixed nitrogen inputs, the aphotic $N_2$ fixation potential of the WTSP remains mostly unexplored (Benavides et al., 2015). Here we quantify $N_2$ fixation in the mesopelagic layer along a ~5000 km transect in the WTSP,

spanning from oligotrophic to ultraoligotrophic conditions (Moutin et al., 2017).

## 2 Materials and methods

### 2.1 Hydrography, nutrients, chlorophyll *a* and dissolved organic carbon

        The OUTPACE cruise (Oligotrophy to Ultraoligotrophy South Pacific Experiment;

http://dx.doi.org/10.17600/15000900) took place onboard the R/V *L'Atalante* from 20 February to 2 April 2015 (i.e. during austral summer), sailing westwards from New Caledonia to French Polynesia (see Figure 2 in Moutin et al., 2017). Temperature, salinity, chlorophyll fluorescence and oxygen data were obtained using a SBE 9plus CTD mounted on a General Oceanics rosette frame fitted with 24 - 12 L Niskin bottles.

Seawater samples were collected with Niskin bottles mounted on a rosette frame at 15 short duration (SD, 8 h) and 3 long duration (LD, 7 days) stations (Moutin et al., 2017). Samples for the



determination of inorganic nutrients nitrate ($NO_3^-$), nitrite ($NO_2^-$) and phosphate ($PO_4^{3-}$)) were collected in 20 mL acid-washed polyethylene flasks, poisoned with 1% mercury chloride, and analyzed onshore using a AA3 Bran+Luebbe autoanalyzer. The detection limit for both $NO_3^-$ and $PO_4^{3-}$ was 0.05 µM.
Samples for the determination of dissolved organic carbon (DOC) were collected in combusted glass

bottles and immediately filtered through two mounted precombusted (4 h, 450ºC) 25 mm GF/F filters (0.7 µm, Whatman) using a custom-made all-glass/teflon filtration syringe system. Filtered seawater was directly collected in precombusted glass ampoules and acidified to pH 2 with orthophosphoric acid. Ampoules were immediately sealed and stored cold (4ºC) and in the dark until analyses by high temperature catalytic oxidation on a Shimadzu TOC-L analyzer according to Sohrin et al. (2005).

Typical analytical precision is ± 0.1 - 0.5 (SD) or 0.2 - 0.5% (CV). Consensus reference materials (http://www.rsmas.miami.edu/groups/biogeochem/CRM.html) were injected every 12 to 17 samples to control for stable operating conditions. Chlorophyll *a* (Chl *a*) concentrations were determined from 500 mL samples filtered through GF/F filters. Chl *a* was extracted in methanol and measured by fluorometry (Herbland et al., 1985).

## 2.2 DOM analysis

Samples for ultra-high resolution mass spectrometry analyses were collected in acid-cleaned 2 L transparent polycarbonate bottles and extracted (solid-phase) via Agilent PPL cartridges as described in Dittmar et al. (2008). After extraction, the cartridges were rinsed with acidified ultrapure water and

frozen at -20°C. Subsequently, the samples were dried by flushing with high-purity $N_2$ and eluted with 6 mL of methanol. The efficiency of the extraction was 47.3 ± 3.9% on a carbon basis. Methanol extracts were molecularly characterized on a 15 Tesla Fourier-transform ion cyclotron resonance mass spectrometer (Solarix FTICRMS) using an electrospray ionization source in negative mode (Bruker Apollo II). Molecular formulae were ascribed to the detected masses as outlined in Seidel et al. (2014).

The aromaticity and unsaturation degree of each compound were evaluated according to its molecular formula and were presented as the modified aromaticity index (AI-mod) and double bond equivalents (DBE), respectively (Koch and Dittmar, 2006). In addition, we ascribed the identified molecular formulae identified to compound groups according to established molar ratios, AI-mod, DBE and heteroatom contents (Seidel et al., 2014). To reveal compositional differences among samples, we

performed a principal coordinate analysis (PCoA) on Bray-Curtis distance matrices, including all detected molecular formulae and their respective relative FTICRMS signal intensities. The PCoA scores were correlated against all hydrographic and biological variables measured in our study.

## 2.3 N$_2$ fixation rates

Seawater was sampled at each SD station from 200, 500, 650 and 800 dbar in quadruplicate 4.3 L transparent polycarbonate bottles covered with black cloth. Each bottle was spiked with 6 mL of $^{15}N_2$



gas (98.9% Euriso-top), inverted 20-30 times, and incubated in the dark at 8ºC in temperature-regulated incubators onboard. After 24 h of incubation, each pair of bottles were filtered onto separate pre-combusted GF/F filters (treated as duplicates), and stored at -20ºC until analyzed with an Integra2 Analyzer, calibrated every ten samples using reference material (IAEA-N1). To obtain accurate $N_2$

5  fixation rates we (1) measured the initial $\delta^{15}N$ of $N_2$ in the incubation on each incubation bottle by membrane inlet mass spectrometry analyses (MIMS; Kana et al., 1994), (2) collected time zero samples in duplicate at each depth and station to determine the natural $\delta^{15}N$ of ambient particulate nitrogen (PN), and (3) subtracted blank GF/F PN values from our results. $N_2$ fixation rates were calculated with the equations of Montoya et al. (1996). Considering the PN linearity limit of the mass spectrometer (2.32

10  µg N), three times the standard deviation of time zero values (natural $\delta^{15}N$ of PN), our usual filtration volume (8.6 L) and incubation time (24 h), our volumetric $N_2$ fixation rate detection limit was 0.027 nmol N $L^{-1}$ $d^{-1}$. The minimum quantifiable rate calculated using standard propagation of errors via the observed variability between replicate samples was 0.006 nmol N $L^{-1}$ $d^{-1}$.

**2.4 Flow cytometry**

Samples for cell enumeration were collected at the same stations and depths as samples for the quantification of $N_2$ fixation rates. Samples of 1.8 mL were fixed (0.25% electron microscopy grade paraformaldehyde, w/v) for 10-15 min at room temperature in the dark, flash-frozen in liquid nitrogen and stored at -80°C for later analysis using a BD Influx flow cytometer (BD Biosciences, San Jose, CA,

USA). Samples were thawed at room temperature, in the dark, and reference beads (Fluoresbrite, YG, 1 µm) were added to each sample. The non-pigmented bacterioplankton (hereafter bacteria) were discriminated in a sample aliquot stained with SYBR Green I DNA dye (1:10,000 final). Because the *Prochlorococcus* population cannot be uniquely distinguished in the SYBR stained samples in the upper water column, bacteria were determined as the difference between the total cell numbers of the SYBR

stained sample and *Prochlorococcus* enumerated in unstained samples. Particles were excited at 488 nm (plus 457 nm for unstained samples) and forward (<15°) scatter (FSC), green fluorescence (530/40 nm), orange fluorescence (580/30 nm) and red fluorescence (>650 nm) emissions were measured. Bacteria were discriminated based on their green fluorescence and FSC characteristics. Cytograms were analyzed using FCS Express 6 Flow Cytometry Software (De Novo Software, CA, US).

**2.5 DNA extraction, sequencing, and sequence analysis**

Samples for DNA extraction were collected in duplicate in 4.3 L polycarbonate bottles and the total volume was filtered immediately through 0.2 µm Supor filters (Pall Gelman). The filters were stored in bead beater tubes and kept at -80ºC until analysis. Samples were collected from four depths

(200, 500, 650, and 850 dbar) at all SD stations, with the exception of station SD5 where seawater for DNA analyses was only collected at 500 dbar, and station SD14 where only 200 and 800 dbar were



analyzed. DNA was extracted using the Qiagen Plant kit, with additional freeze-thaw, bead beating, and

Proteinase K steps for sample preparation before the kit purification, and elution to 100 μL as

previously described (Moisander et al., 2008). PCR was conducted using degenerate, nested *nifH*

primers (Zehr et al., 2001), and the second round primers modified for Illumina library preparation,

using a Bio-Rad C1000 thermocycler. The PCR mix was composed of 2.5 μL of 10X Platinum Taq

PCR buffer (Thermo Fisher), MgCl (2.5 mM final), dNTPs (0.2 mM final), primers (1 μM final), 0.11

μL Platinum Taq, and 4 μL of DNA extract (1 μL on second round), adjusted to 25 μL total volume

with nuclease free water. To alleviate PCR biases, PCR was conducted in triplicate in each round of

reactions where first round triplicate reaction products were pooled, then 1 μL was used as a template in

triplicate reactions on the second round. A negative PCR control with water as template was included

and treated in parallel with samples through all the subsequent steps. Sub-samples of the amplification

products were checked via 1.2% TAE gel electrophoresis. The second round products showing a band

of the expected size were pooled and purified using a magnetic bead protocol (Ampure, Beckman

Coulter). The purified products were barcoded for Illumina (San Diego, CA, USA) MiSeq sequencing

with Nextera indexes, using the manufacturer's protocol. The indexed products were purified again with

magnetic beads, then quantified with a plate reader (Molecular Devices, Sunnyvale, CA, USA) using

Picogreen (Thermo Fisher). The indexed samples were adjusted to equal concentrations and pooled for

multiplexing during sequencing. The pooled sample was shipped to the Tufts University sequencing

center (Boston, MA, USA) for paired end sequencing (2x300). The quality of the pooled sample and

20 select individual samples was checked with a Bioanalyzer before the run. The resulting sequences were

paired within Mothur (Schloss et al., 2009), and reads containing ambiguities or more than 8

homopolymers were discarded. The sequences were assigned to OTUs (at 97% cutoff) using the Uclust

denovo picking method (Edgar, 2010) implemented in MacQIIME v1.9.1 (Caporaso et al., 2010). Low

abundance OTUs consisting of less than 15 sequences across all samples were discarded from further

processing. A representative sequence of each OTU was extracted from the data and quality processed

in Arb (Ludwig et al., 2004), removing sequences that did not conceptually translate or were otherwise

of poor quality. These OTUs were removed from further analysis. Remaining sequences were aligned

based on protein alignment of the *nifH* fragments in a public *nifH* database

(https://www.jzehrlab.com/nifh). Aligned protein sequences were assigned to *nifH* clusters using a

30 decision tree statistical model, CART (Frank et al., 2016). The sequence data were normalized to

proportion of total reads in each sample. Total relative abundances of sequences that fell in major *nifH*

clusters were used to create a heatmap within R Studio using the VEGAN statistical package (Oksanen

et al., 2015). Sequences were further classified through a locally run blastp using the *nifH* database as a

reference (April 2015 database update). A neighbor joining tree was built in Arb with the 100 most

35 abundant OTUs across all samples.





## 3 Results

### 3.1 Hydrography, nutrients, DOC and bacterial abundance

Sections of hydrographic variables (temperature, salinity), nutrients ($NO_x$ -i.e. $NO_3^-$+$NO_2^-$- and

$PO_4^{3-}$), DOC and bacterial abundance are shown in Fig. S1. All variables show a clear divide between the Melanesian Archipelago waters (MA; stations SD1 to SD12) and the South Pacific Gyre (GY; eastwards of SD12) (Moutin et al., 2017). Lower temperatures and salinity values (<8ºC and ~35, respectively) were measured below 450 to 600 dbar in the MA, while they were detected at shallower depths eastwards in the GY (Figs. S1a, b). Nutrient concentrations were high throughout the

mesopelagic zone west of New Caledonia (>30 µM $NO_x$ and >2 µM $PO_4^{3-}$; and Figs. S1c, d). Sailing eastwards, $NO_x$ in the MA region was <5 µM between 150 and 250 dbar, with high concentrations >30 µM being detected at ~680 dbar. Such high $NO_x$ concentrations were detected at shallower depths (500 to 600 dbar) in the GY. $PO_4^{3-}$ followed a similar pattern, with the highest concentrations detected at depths >500 dbar reaching 2.5 µM.

DOC concentrations presented a pattern opposed to that of inorganic nutrients, with the <300 dbar presenting concentrations 50 - 60 µM, lowering to <40 µM below 600 dbar (Fig. S1e). Bacterial abundance was >1 x $10^5$ cells $mL^{-1}$ down to 300 dbar in the MA waters, while its numbers decreased abruptly in the GY, especially east of SD12 (Fig. S1f)

### 3.2 High-resolution analysis of DOM (FTICRMS)

FTICRMS analysis of DOM yielded ~13500 molecular formulae in each sample, covering a mass range between 150 and 1000 Da. Each molecular formula was assigned to a given compound group as described in Seidel et al. (2014). According to this grouping, 4-31% of all compounds detected were oxygen-poor (O/C<0.5) unsaturated aliphatics, 16% were oxygen-rich (O/C>0.5) unsaturated

aliphatics, 13% were polyphenols, while saturated fatty acids, sugars and peptides represented <3%. Compounds usually regarded as labile DOM (peptides, sugars and saturated fatty acids) were relatively more abundant in the MA (data not shown).

### 3.3 Aphotic $N_2$ fixation rates

Aphotic $N_2$ fixation rates were measureable at all stations and depths and ranged between 0.05 and 0.68 nmol N $L^{-1}$ $d^{-1}$ (Fig. 1). These rates did not seem to follow any longitudinal or vertical pattern. However, the rates observed at station 13 (where a massive surface concentration of chlorophyll was observed; de Verneil et al., 2017) were on average ~5-fold higher than at the other stations (average $0.63 \pm 0.07$ nmol N $L^{-1}$ $d^{-1}$).

### 3.4 Diazotroph community composition



There were 3317-146864 (mean = 88867, s.d. = 42574) sequences per sample after pairing and the QA/QC steps. The negative control resulted in only 8 reads belonging to 6 OTUs. These sequences are likely a result of misbinning at the time of sequencing, and therefore the negative control was removed from downstream analysis. Shannon diversity at the 97% OTU level was not significantly

affected by either depth or station (one-way ANOVA p>0.05).

The majority of *nifH* sequences fell to Clusters 1 and 3, although Clusters 2 and 4 were represented in the transect at low relative abundances (Fig. 2). Within Cluster 1, subcluster 1G, which contains Gammaproteobacteria, accounted for over half of the total sequences (56%, s.d. = 38%). With a few exceptions, in samples with lower proportions of 1G, subcluster 3S was the most abundant group.

Cluster 3S had high relative abundances at station SD10 and in the 200 and 500 dbar depths of stations SD2 and SD12, as well as at 200 dbar at station SD13. The cyanobacterial subcluster 1B was observed at very low relative abundance throughout all stations and depths (average 0.5% of total community) and included *Trichodesmium*, UCYN-A, and *Richelia*. Consistent variations in community composition among depths and stations were not detected via cluster-based analysis, however, patterns emerged

when observing data at the OTU level in abundant clusters (Fig. 3). In subcluster 1G, reads most closely matching an unclassified bacterium from the tropical North Atlantic (Unc12217, E value = $2.82 \times 10^{-70}$; Table 1, Fig. 4) in the *nifH* database dominated the communities in the 650 dbar depth, and this phylotype was found only at minor proportions in other depths. This phylotype had an approximately 97% amino acid identity with *Vibrio diazotrophicus* (Fig. 4). This Unc12217/*V. diazotrophicus* related

phylotype was present at high proportions across all other 650 dbar samples except at station SD2, the westernmost station. Although identified as "Other 1G" in Fig. 3, this trend was also true for several other groups present only at the 650 dbar depth (best matches with database sequences Unc12243, Unc12270, and UncPr491; Table 1). An additional group was found only at 650 dbar in the easternmost stations (SD10-SD13), closely matching a sequence from the Amazon River (UncM2163, E value =

$4.22 \times 10^{-74}$; Table 1). Within the *nifH* Cluster 3, subcluster 3S was the most widespread and abundant, with representation from three groups in the reference database: Unc12045, UncB2403, and UncMa132. Sequences with best matches with Unc12045 (Genbank ID: ADV51583; Turk et al., 2011), and UncB2403 (AAP48957; Steward et al., 2004) were present at stations SD2, SD4, SD10, and SD12, but had the highest relative abundance at station SD10, 500 dbar at station SD2, and the 200 and 500 dbar

depths of station sd12. UncMa132 (AAS98182; unpublished) related groups were recovered from all stations at a low relative abundance, with the highest relative abundance in the 200 dbar samples from stations SD2, SD10, SD12, and SD13, and the 800 dbar samples from stations SD4 and SD10. These 3S groups have no closely related cultivated isolates, with the closest similarity with *Spirochaeta aurantia* (74-78% similarity, AF325792). All of the top 100 most abundant OTUs fell in Clusters 1 and 3 (Fig.

4). The majority fell in the Cluster 1G, with several additional phylotypes present in addition to the major ones discussed above. Several OTUs were closely related with previously described sequences



from the South Pacific Ocean mesopelagic layers (Benavides et al. 2015). *Magnetococcus* sp. and *Methylomonas* sp., and *Teredinibacter turnerae* were among closest cultivated representatives to Cluster 1G OTUs recovered (Fig. 4). Among the OTUs that fell in the Cluster 3, *Desulfovibrio* spp. were the closest cultivated representatives in the NCBI database.

**3.5 N$_2$ fixation and diazotrophs related to in situ environmental parameters**

Bonferroni-corrected Spearman rank correlations showed that N$_2$ fixation rates were only significantly correlated with temperature ($\rho = 0.263$, p = 0.045), salinity ($\rho = 0.284$, p = 0.029), and DOC concentrations ($\rho = 0.269$, p = 0.042; note that these hydrographic variables and DOC were intercorrelated between them, data not shown; Table S1). A redundancy analysis (RDA; Fig. 5) including hydrographic variables, inorganic nutrients, DOC, bacterial abundance, N$_2$ fixation rates and DOM PCoA scores indicated that N$_2$ fixation rates were not related to DOM compositional variability (Table S1). Shallower samples such as those from 200 dbar were significantly related to temperature, salinity, oxygen and DOC concentrations, while the first principal coordinate of DOM related to the majority of the samples. Stations SD13 and SD15 (easternmost part of the transect, within the GY) were partly related to depth and NOx concentrations (samples from 650 and 800 dbar), while the rest of the samples of the profile appeared very distant in the RDA plot (Fig. 5).

**4 Discussion**

The aphotic N$_2$ fixation activity measured in the WTSP was low (average $0.18 \pm 0.07$ nmol N L$^{-1}$ d$^{-1}$), but consistently detected across all depths and stations (Fig. 1), representing on average 13 and 51% of photic N$_2$ fixation, in the MA and GY waters, respectively (Bonnet et al., submitted to this issue). It is pertinent to note that aphotic N$_2$ fixation rates may be underestimated if a significant percentage of the non-cyanobacterial diazotroph population is smaller than 0.7 µm (the nominal pore size of GF/F filters), as N$_2$ fixation rates in non-cyanobacteria dominated environments is significantly higher when smaller pore size filters are used (Bombar et al., submitted). Aphotic N$_2$ fixation may contribute significantly to global fixed nitrogen inputs if widespread throughout the ocean's mesopelagic zone (or deeper). Unfortunately, our ability to assess this contribution remains hindered by the lack of specific N$_2$ fixation methods and our poor understanding of the ecophysiology of non-cyanobacterial diazotrophs (Bombar et al., 2016; Moisander et al., 2017).

N$_2$ fixation rates in aphotic environments correlate with different DOM compound groups in different regions (Benavides et al., 2015; 2016), and *nifH* gene expression varies among non-cyanobacterial diazotroph phylotypes when exposed to conditions presumed to enhance their N$_2$ fixation activity (Severin et al., 2015). N$_2$ fixation was detected as low rates across an oligotrophic-to-ultraoligotrophic transect in the WTSP. These rates were not significantly correlated to DOM compounds as identified by FTICRMS, despite they were positively correlated to DOC concentrations.





Non-cyanobacterial diazotroph communities are usually highly diverse in aphotic marine waters (e.g. Hewson et al., 2007). If such phylogenetic diversity also entails a broad metabolic diversity and different affinities for DOM compounds, correlations between DOM and non-cyanobacterial diazotroph abundance, identity and/or $N_2$ fixation activity will likely be blurred. Such ecophysiological

heterogeneity may be also be reflected by the lack of longitudinal or vertical patterns in $N_2$ fixation rates observed along the transect (Fig. 1).

Some depth- and longitude related patterns were observed within the potential diazotroph community, one of the major patterns being a distinct *V. diazotrophicus* related phylotype from the major *nifH* subcluster 1G dominating at the 650 dbar depth. These sequences were unique from the 1G

sequences found at other depths, and, with the exception of station SD2, were found uniformly across stations. A potential cause for the depth variation seen at 650 dbar is the presence of oxygenated Sub-Antarctic Mode Water (SAMW) at this depth (Fumenia et al., submitted to this issue). The high concentration of oxygen in this water mass (190-220 µmol kg$^{-1}$) could potentially shift the diazotroph community to members that can withstand higher levels of oxygen. To our knowledge, this is the first

study identifying a relationship between *nifH* community composition and large-scale oceanographic circulation patterns in mesopelagic depths.

Members of the 1G subcluster include a variety of Gammaproteobacteria, and this group has previously been reported at high numbers in tropical surface waters including in the WTSP (Messer et al., 2017). In mesopelagic waters, transcripts from the 1G subcluster have been reported in an oxygen-

deficient zone in the Arabian Sea at a depth of 175 m (Jayakumar et al., 2012), and genes have been reported from the WTSP from depths of 350-600 m (Benavides et al., 2015). Closely related cultivated isolates to the sequences found in this study include members of the Gammaproteobacterial genera *Vibrio, Pseudomonas, Klebsiella*, and *Agrobacterium*. The cyanobacterium *Pseudanabaena* also had relatively close relationship to the 650 dbar subcluster 1G sequences. When the proportion of sequences

from subcluster 1G was low, members of cluster 3, primarily subcluster 3S, tended to have higher relative abundances (mostly east of the Tonga Trench, located between stations SD9 and SD10; Fig. S1). The three major matches of sequences in this study within the Arb database for subcluster 3S were of sequences reported from the surface waters in the North Pacific Subtropical Gyre and the Eastern North Atlantic, and a hypersaline lake. The phylotype most commonly observed at 200 dbar was most

similar to sequences reported in the tropical North Pacific (AAS98182). Cluster 3 is typically considered to contain obligate and facultative anaerobes including *Spirochaeta* and *Desulfovibrio*. Cluster 3 diazotrophs were present in low gene copy numbers in North Atlantic surface waters, even when $NO_3^-$ concentrations were high (Langlois et al., 2008). Members of Cluster 3 have also been recovered in the mesopelagic WTSP (Benavides et al., 2015), although this study reports longitudinal

variation as a primary driver of Cluster 3 phylotype diversity and relative abundance.





The cyanobacterial subcluster 1B including *Trichodesmium*, UCYN-A, and *Richelia* was observed at very low relative abundance throughout all stations and depths (average 0.5% of total community), in agreement with the findings of Caffin et al. (2017), who detected those phylotypes in sediment traps deployed during the OUTPACE cruise at 150 and 325 m. Dead *Trichodesmium* colonies are thought to be mainly degraded in the photic zone (Letelier and Karl, 1998), although the detection of *Trichodesmium* in sediment traps and seawater samples obtained from the mesopelagic zone (Agustí et al., 2015; Chen et al., 2003) suggests that decayed dense blooms likely sink fast down the water column. The detection of cyanobacterial diazotroph *nifH* sequences in the mesopelagic zone questions whether the $N_2$ fixation rates measured are solely attributable to non-cyanobacterial diazotrophs (Moisander et al., 2017). Cyanobacterial photosynthetic diazotrophs reach the mesopelagic zone through sinking and sedimentation, and thus are unlikely diazotrophically active when devoid of light. However, recent studies have detected photosynthetically active diatoms at depths overpassing the mesopelagic zone (Agustí et al., 2015), indicating that dead cell packages can be exported vertically at high speed. If cyanobacterial diazotrophs remain active when they reach the aphotic layer, or if they die or shut down $N_2$ fixation on the way, remains an open question.

The data presented here are a significant contribution to the scarce availability of aphotic $N_2$ fixation rates, generally ocean wide, and specifically in the WTSP. Despite our knowledge on the ecophysiology of aphotic non-cyanobacterial diazotrophs is limited (Bombar et al., 2016), their ubiquity in the mesopelagic layer and the consistent detection of active $N_2$ fixation activity at all depths sampled during our study suggest that aphotic $N_2$ fixation may contribute significantly to fixed nitrogen inputs in this area.

**Acknowledgements**

This is a contribution of the OUTPACE (Oligotrophy from Ultra-oligoTrophy PACific Experiment) project (https://outpace.mio.univ-amu.fr/) funded by the French research national agency (ANR-14-CE01-0007-01), the LEFE-CYBER program (CNRS-INSU), the GOPS program (IRD), and the CNES (BC T23, ZBC 4500048836). The OUTPACE cruise (https://doi.org/10.17600/15000900) was managed by MIO (OSU Institut Pythéas, AMU) from Marseilles (France). The authors thank the crew of the R/V *L'Atalante* for outstanding on-ship operations. M.B. was funded by the People Programme (Marie Skłodowska-Curie Actions) of the European Union's Seventh Framework Programme (FP7/2007-2013) under REA grant agreement number 625185. NSF OCE-1733610 award to P.M. supported P.M. and K.S. S.D. was funded by the NSF award OCE-1434916 and received institutional support funded by the Vetlesen Foundation.

35



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



**Tables**

**Table 1: Blastp identities to members of the 1G subcluster.**

| *nifH* ID | Accession number | Study | Location reported | Closest cultivated relative (Identity, E value) |
|---|---|---|---|---|
| UncM2172 | ABV00657 | Unpublished (Moisander/Subramaniam et al.) | Gulf of Guinea | *Vibrio diazotrophicus* (98%, 1e-71) |
| Unc12217 | ADV51755 | Turk et al. (2011) | Tropical Eastern N. Atlantic | *Agrobacterium tumefaciens* (83%, 4e-55) |
| Unc18425 | BAN66776 | Unpublished (Shiozaki et al.) | Northern S. China Sea | *Pseudomonas stutzeri* (86%, 1e-64) |
| Unc17727 | ABV00657 | Unpublished (Moisander/Subramaniam et al.) | Gulf of Guinea | *Klebsiella pneumoniae* (100%, 4e-95) |
| Unc11967 | ADV35118 | Unpublished (Olson and Lesser) | Florida Keys | *Pseudanabaena cf. persicina* (97%, 3e-78) |
| UncMa745 | ABX39720 | Moisander et al. (2008) | S. China Sea | *Pseudomonas stutzeri* (96%, 1e-70) |
| UncPr802 | AEA49463 | Fernandez et al. (2011) | Eastern S. Pacific | *Vibrio natriegens* (96%, 3e-74) |
| UncMa806 | AAY60084 | Langlois et al. (2005) | Atlantic Ocean | *Agrobacterium tumefaciens* (96%, 1e-70) |
| UncM2163 | ABF21183 | Unpublished (Hewson and Fuhrman) | Amazon River | *Vibrio diazotrophicus* (81%, 6e-55) |
| Unc12136 | ADV51674 | Turk et al. (2011) | Tropical Eastern N. Atlantic | *Agrobacterium tumefaciens* (96%, 2e-69) |
| Unc12270 | ADV51808 | Turk et al. (2011) | Tropical Eastern N. Atlantic | *Vibrio diazotrophicus* (79%, 1e-52) |
| UncMa747 | ABX39731 | Moisander et al. (2008) | S. China Sea | *Pseudomonas stutzeri* (84%, 6e-60) |
| Unc12551 | ADO20633 | Halm et al. (2012) | S. Pacific Gyre | *Pseudanabaena cf. persicina* (95%, 2e-70) |
| UncMa832 | ABD62932 | Unpublished (Foster et al.) | Atlantic Ocean | *Pseudomonas stutzeri* (84%, 5e-60) |
| Unc15356 | AER93057 | Unpublished (Lopez) | Mexican oasis system soil | *Pseudanabaena cf. persicina* (95%, 1e-78) |
| UncPr491 | AEA49150 | Fernandez et al. (2011) | Eastern S. Pacific | *Pseudanabaena cf. persicina* (98%, 3e-76) |
| Unc12243 | ADV51781 | Turk et al. (2011) | Tropical Eastern N. Atlantic | *Vibrio diazotrophicus* (80%, 2e-54) |





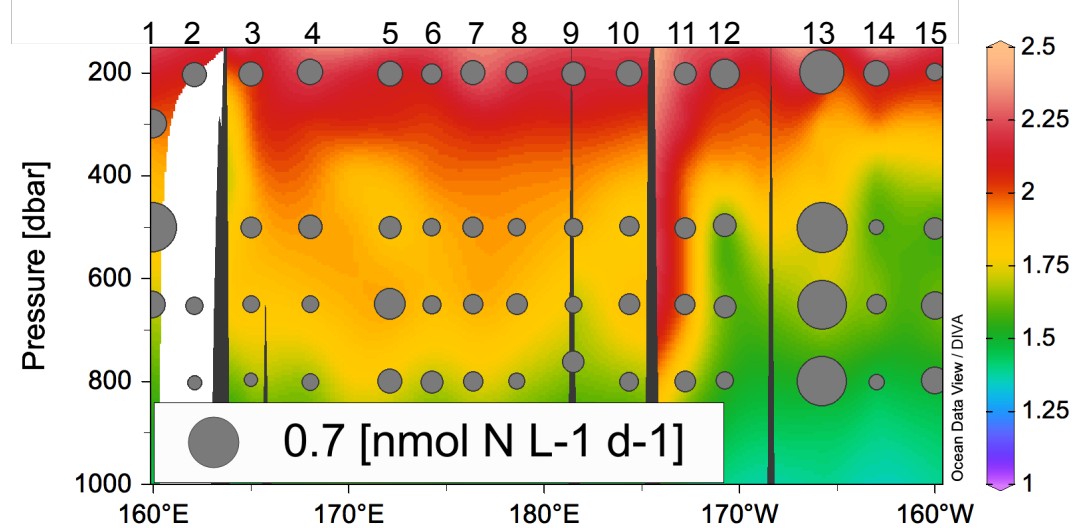

**Figure 1: Longitudinal section of N₂ fixation (nmol N L⁻¹ d⁻¹, as sized circles, reference size shown on the bottom left of the panel) superimposed on dissolved oxygen concentrations (color scale; mL L⁻¹). For reference, the station numbers are displayed on top of the panel.**





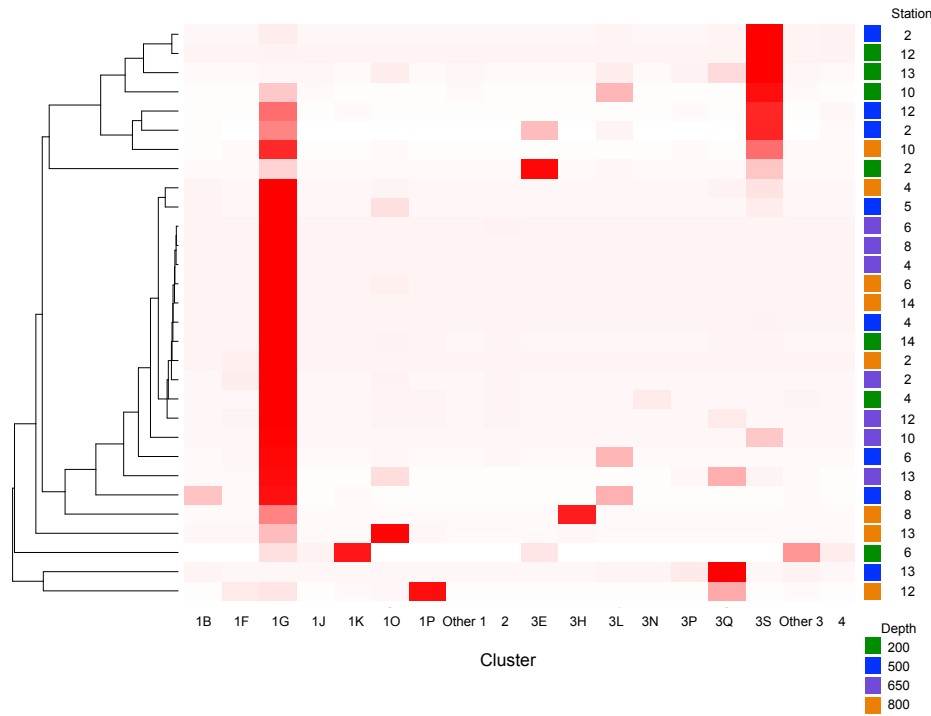

**Figure 2: Heatmap of *nifH* clusters and subclusters across the stations. The Bray-Curtis distances were used to build the dendrogram on the left. Distances were calculated with relative abundances of sequences by subcluster. Subclusters within Clusters 1 and 3 that had low relative abundances throughout all samples have been grouped as "other".**



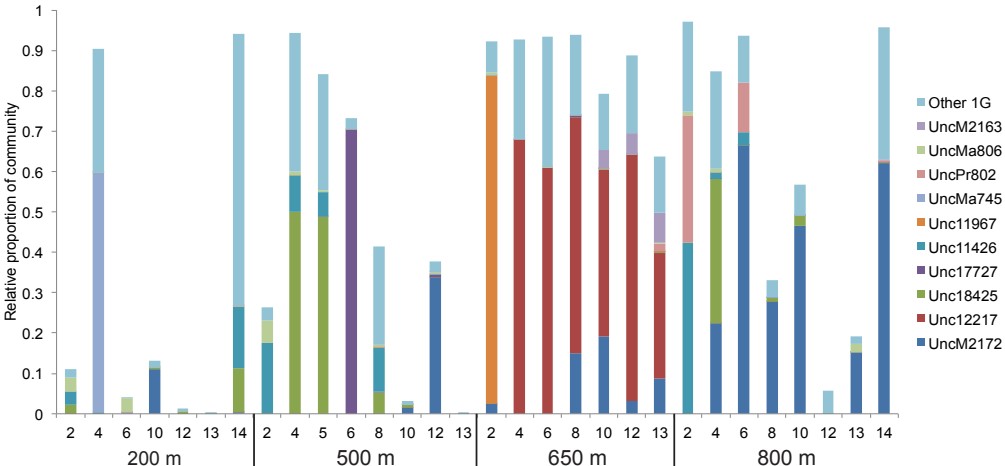

Figure 3: Relative abundance of subcluster 1G over depths and stations. Each bar represents the
relative abundance of subcluster 1G in the total community for each sample. Samples are

5  arranged by station within each of the depths sampled. Blastp was used to assign OTUs to a top
hit in the *nifH* reference database. The major sequence types in the *nifH* database found in the
subcluster 1G are shown with different colors.



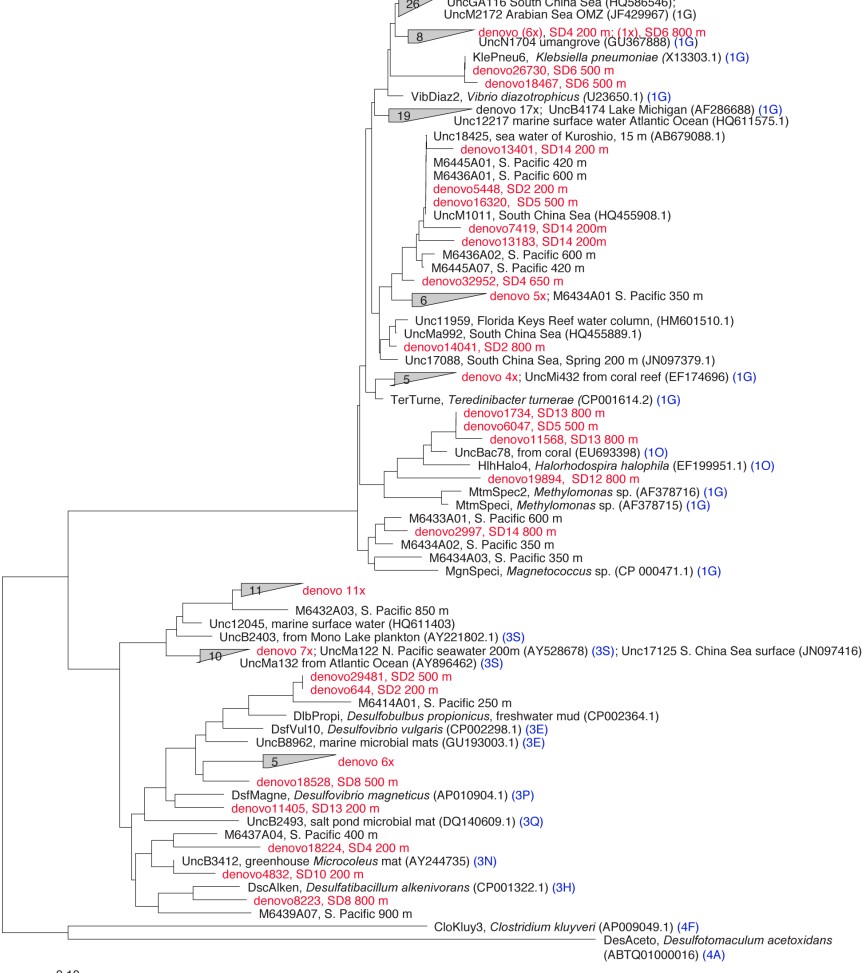

**Figure 4: A *nifH* amino acid neighbor-joining tree with 100 most abundant OTUs from this study shown. Sequences 'denovo-' shown in red are randomly chosen representative sequences from these OTUs (OTUs binned at 97% identity). The clusters in the tree are grouped at an approximately >95% sequence identity. The tree includes reference sequences (if uncultivated, names of these sequences start with 'Unc'). The reference sequences are shown with accession numbers from the *nifH* database and with the cluster identifier shown in blue if indicated in the *nifH* database. Additional sequences are included from a previous study in the South Pacific mesopelagic layers (Benavides et al. 2015); these clones are labeled with the original clone names M64XXAXX and depths.**



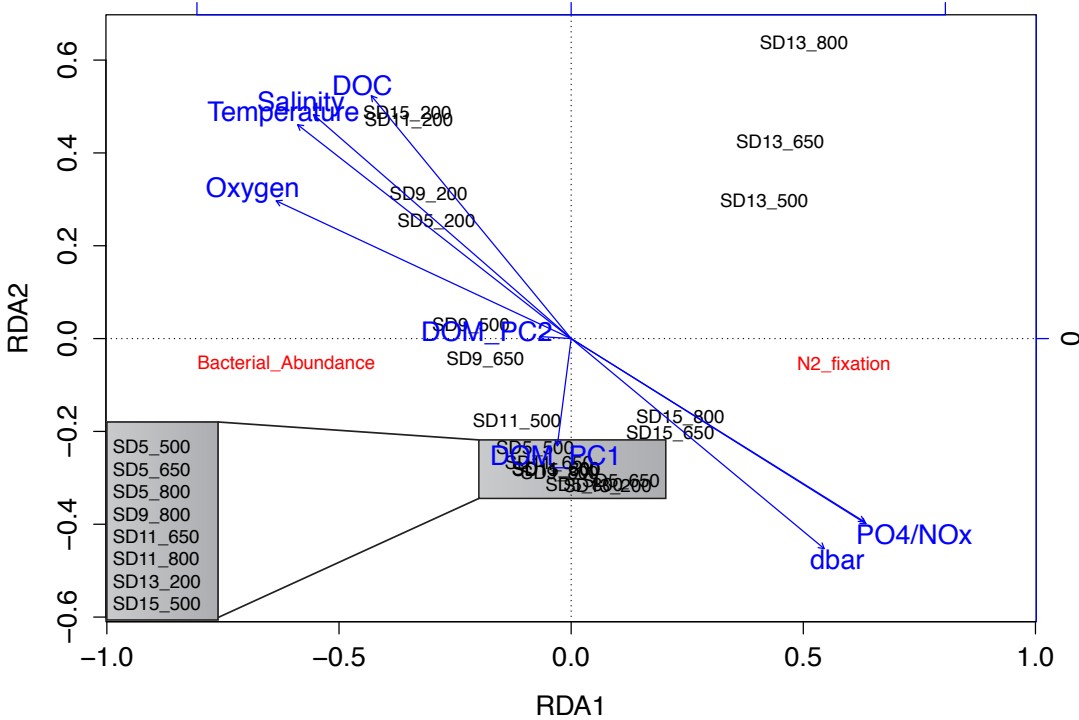

**Figure 5: Redundancy analysis (RDA) ordination biplot showing the relationship between N$_2$ fixation rates, bacterial abundance, depth, environmental variables (temperature, salinity and oxygen), dissolved organic matter (DOM) principal coordinates, dissolved organic carbon (DOC) and inorganic nutrient concentrations.**