# Peer review of "Aphotic N2 fixation along an oligotrophic to ultraoligotrophic transect in the Western Tropical South Pacific Ocean"

_Biogeosciences, 2017_

## Referee Comment (RC1) · L. I. F. Falcon (Referee) · 22 Jan 2018

In this research paper, Benavides et al., make a very solid contribution to current knowledge of N2 fixation rates, N2-fixer diversity, and overall parameters associated to aphotic regions of the oceans. The research presented is excellent in all its aspects-introduction, MM, results and discussion are solid, straightforward, and present data produced with state-of-the art technologies. Although local N2 fixation rates are low compared to some photic environments, they were constantly found along the sampling gradient reported, and integrated rates suggested in this research can consist of up to ~50% of surface rates. I believe this is one of the main contributions of this

research. A second main contribution is the finding of V. diazotrophicus related phylotypes dominating at the 650 dbar depth, associated to SAMW- which suggests a relationship between nifH diversity and a large-scale water mass. Further, the results on Trichodesmium diversity found in the aphotic layers, suggesting the effect of large cyanobacterial bloom sinking, is also very relevant to understand the N cycle.

———————————————

---

## Referee Comment (RC2) · Anonymous Referee #2 · 31 Jan 2018

The manuscript submitted by Benavides et al reports on rates of aphotic nitrogen fixation in the wester tropical South Pacific Ocean. In parallel, the group try to identify the diazotrophs present at depth and also the environmental factors supporting aphotic diazotrophy. The manuscript is well written and the investigation is mostly thorough, as it should be in reporting such low rates of nitrogen fixation. Aphotix diazotrophy is an emerging story that has yet to be reconciled completely in terms of its significance. This manuscript provides new data that will add to this emerging story. The manuscript is certainly relevant to the Biogeosciences community. I have some suggestions and concerns that should be addressed prior to publication:

[Figure]

1. Your suggestion that the nif genes are associated with a water mass are interesting. Can you show this using a T/S plot with your 'z' value being either nif gene or a measure of the diazotroph community? Do you see higher rates here too? In figure 1, there are higher rates = larger dots at ~ 165W. Is this the same station/region where you see high V. diazotrophicus? If you plot n2 fixation rates on a T/S plot, are there any patterns with water masses?

2. I am not sure what the high resolution analysis of DOM by FTICRMS adds to this manuscript. As stated in the abstract and on page 9, line 10, the n2 fixation rates were not related to DOM compounds analysed by FTICRMS. The application of such techniques may have been more suitable in an incubation-type experiment, e.g. adding compounds and detecting their uptake and/or incorporation.

3. Why would fixed N inputs add to this area only if diazotrophy is related to water masses which are moving around the ocean? Is this really only a locally important processes add N to this area only?

4. Unclear why the depth is reported as dbar here. I suggest the authors change dbar to meters.

Figure 1. I suggest that oxygen is reported as umol L-1 or umol kg-1 and not mL L-1 which is an unconventional unit for oxygen on oceanography. This figures is not clear because it is not possible to see the specific rates of nitrogen fixation here. I suggest this is replotted to show the actual values for nitrogen fixation, which would be more useful considering the uniqueness of this data set.

Figure 5. This is not clear due to words in blue overlapping as well as SD5 to SD15 overlapping. Can this be replotted, e.g as colour codes?

Figure S1. The DIN and phosphate around station 7 look odd? There is no DIN and phosphate between 400 and 1000m. Please check.

Table S1 has fallen off the bottom of the page. Please explain in the legend how to

interpret the numbers. Are these p values or is a high value good, i.e. means a strong relationship. What do the stars mean?

Minor details/comments: Abstract, line 29: remove 'here'. Change of tense, suggest 'we measured....and identified...'

This sentence is awkward 'Because non-cyanobacterial diazotrophs presumably need external dissolved organic matter (DOM) sources for their nutrition, we also identified DOM compounds using Fourier Transform Ion Cyclotron Mass Spectrometry (FTICRMS)' - suggest change to 'DOM sources were identified.....because non-cyans...

Page 2, line 1: remove majorly

Page 2, line 8: '....that aphotic N2 fixation may contribute significantly to fixed nitrogen inputs in this area.' As above....Why just this area? Considering the deep ocean consists of water masses moving water and its properties around the ocean, what would the nitrogen fixation here contribute to the N budget here only?

Page 3: Line 17: the N2 fixation rate should be removed as a volumetric rate rather than integrated rate. For example, it may only be high because it is integrated over a thick layer of the ocean?

Page 5: Line 5: 'measured the initial $\delta$15N of N2 in the incubation on each incubation bottle by membrane inlet mass spectrometry analyses (MIMS; Kana et al., 1994)' - do you mean after the addition of 15N2? Then this needs to be clearer here. But range of enrichments were you achieving here? In light of the newness of this approach, it would be appropriate to include some detail here.

Page 11: Note that Tricho colonies have been detected in sediment traps elsewhere, e.g. Pabortsava et al 2017 in Nature Geosciences

---

## Referee Comment (RC3) · Anonymous Referee #3 · 6 Feb 2018

The study by Benavides and coauthors report aphotic N2 fixation rates and identify diazotrophs present in the mesopelagic layer of the western tropical South Pacific. The paper is a significant contribution which increases the knowledge about aphotic nitrogen fixation in a region which is highly interesting in terms of N-input from N2 fixation. Rates of N2 fixation were low but detected across all depths and stations. Shifts in diazotroph assemblages seemed to be mostly associated with depth. A distinct 1G phylotype was identified to coincide with the oxygenated Sub-Antarctic Mode Water. The paper is very well written and the methods used are well described, solid and established.

In my opinion the presentation of data could be improved by clearer links to different water masses. In the title the oligotrophic to ultraoligotrophic transect is highlighted but the way that this translates into sampling stations and different water masses is not evident to the reader from the figures. Further the nifH data is presented largely based on depth rather than sampling location/water mass.

The DOM analysis is valid but considering the low abundances these diazotroph groups are likely present in compared to other members of the microbial community establishing connections may be difficult. From the results section is not evident if differences in DOM compounds were seen across the transect or different depths.

The (relatively) high N2 fixation rates at station 13 are curious and could be given some more attention in the discussion. From Fig 2. it appears like the diazotroph composition from station 13 differs largely between the depths and clusters away from the other samples. I find it very intriguing that this suggests that several different groups may be responsible for similar rates at the different depths. It is mentioned that high concentrations of chlorophyll were observed at this station. Did this coincide with high photic N2 fixation rates?

Other comments: The presentation of average N2 fixation rates and relation to % of photic N2 fixation is unclear and values in abstract and text appear to be different. (Abstract Lines 33-34 and Discussion Lines 20-23)

The Bray-Curtis distances in Figure 2 might be more meaningful if done on a level with higher resolution. Currently the variations in phylotypes is largely "hidden" in the 1G subcluster. A rarefaction to equal sampling depth would further improve this analysis.

In Figs. 2 and 3 data is presented as depth but in Fig. 1 as pressure [dbar]

Fig. 1 Please adjust the scale so that the circles are not cut for stations 1 and 15

---

## Author Response (AR1)

L. I. F. Falcon (Referee)
falcon@ecologia.unam.mx

In this research paper, Benavides et al., make a very solid contribution to current knowledge of N2 fixation rates, N2-fixer diversity, and overall parameters associated to aphotic regions of the oceans. The research presented is excellent in all its aspects-introduction, MM, results and discussion are solid, straightforward, and present data produced with state-of-the art technologies. Although local N2 fixation rates are low compared to some photic environments, they were constantly found along the sampling gradient reported, and integrated rates suggested in this research can consist of up to ∼50% of surface rates. I believe this is one of the main contributions of this research. A second main contribution is the finding of V. diazotrophicus related phylotypes dominating at the 650 dbar depth, associated to SAMW- which suggests a relationship between nifH diversity and a large-scale water mass. Further, the results on Trichodesmium diversity found in the aphotic layers, suggesting the effect of large cyanobacterial bloom sinking, is also very relevant to understand the N cycle.

*We are greatly pleased by this reviewer's positive comments and appreciation of our work. We note that there was an error in our text, and in fact, as shown in Table 1, Unc12217 is most closely related with Agrobacterium tumefaciens (83%), not V. diazotrophicus. The OTU still falls within Gammaproteobacteria and near the Vibrio spp. cluster. To maintain consistency with Table 1, we have changed the text in the revised manuscript. Overall this change does not affect our conclusions.*

**Reviewer #2**
The manuscript submitted by Benavides et al reports on rates of aphotic nitrogen fixation in the wester tropical South Pacific Ocean. In parallel, the group try to identify the diazotrophs present at depth and also the environmental factors supporting aphotic diazotrophy. The manuscript is well written and the investigation is mostly thorough, as it should be in reporting such low rates of nitrogen fixation. Aphotix diazotrophy is an emerging story that has yet to be reconciled completely in terms of its significance. This manuscript provides new data that will add to this emerging story. The manuscript is certainly relevant to the Biogeosciences community. I have some suggestions and concerns that should be addressed prior to publication:

1. Your suggestion that the nif genes are associated with a water mass are interesting. Can you show this using a T/S plot with your 'z' value being either nif gene or a measure of the diazotroph community? Do you see higher rates here too? In figure 1, there are higher rates = larger dots at ~165W. Is this the same station/region where you see high V. diazotrophicus? If you plot n2 fixation rates on a T/S plot, are there any patterns with water masses?

*In this study, our objective was to sample throughout the mesopelagic zone, not necessarily targeting any specific water masses. The depths sampled (200, 500, 650 and 800 db) were "arbitrarily" chosen according to water volume availability in deep casts during the OUTPACE cruise (note that we needed as much as 40 L per depth to perform all our analyses). Very interestingly, when examining the nifH sequencing results, it turned out that a specific phylotype was predominant in a given water mass (subcluster 1G in the SAMW). Unfortunately, the coverage of our samples throughout the mesopelagic zone is not enough to represent all the different water masses present and to identify patterns in N2 fixation activity or diversity of diazotrophs according to water mass distribution.*

*This can be clearly seen in the T-S diagrams shown in Figure 1 (from the response to reviewers file). On the left, we present a T-S diagram of the water masses sampled during the OUTPACE cruise (as displayed in Fig. 4a in Fumenia et al., this issue). According to this T-S diagram, our N2 fixation and nifH gene measurements (central and right figures) correspond to the lower part of the upper thermocline ($\sigma_t$=24.7-25.4), lower part of the lower thermocline ($\sigma_t$=26.5-26.7), and SAMW/AAIW ($\sigma_t$=26.7-27.3). No measurements are available in the two water masses of the central thermocline.*

[Figure]

2. I am not sure what the high resolution analysis of DOM by FTICRMS adds to this manuscript. As stated in the abstract and on page 9, line 10, the n2 fixation rates were not related to DOM compounds analysed by FTICRMS. The application of such techniques may have been more suitable in an incubation-type experiment, e.g. adding compounds and detecting their uptake and/or incorporation.

*Our group investigated aphotic N2 fixation and its relationship with DOM during two cruises in the Solomon Sea (Benavides et al., 2015) and in the Mediterranean Sea (Benavides et al., 2016), in the frame of the project DIADOM https://cordis.europa.eu/project/rcn/187917_en.html*

*In both cases we found positive correlations between labile compounds and N2 fixation. In the OUTPACE cruise we basically followed the same sampling strategy, but did not find significant relationships between DOM composition and aphotic N2 fixation. Although the FTICRMS data may itself not add much to the present study,, we decided to keep it for comparison with our previous studies and to reinforce the need for a mechanistic understanding of how non-cyanobacterial diazotrophs interact with DOM in the ocean.*

*Benavides, M., H. Moisander, P., Berthelot, H., Dittmar, T., Grosso, O. and Bonnet, S.: Mesopelagic N2 fixation related to organic matter composition in the Solomon and Bismarck Seas (Southwest Pacific), PLoS One, 10(12), 1–19, doi:10.1371/journal.pone.0143775, 2015.*

*Benavides, M., Bonnet, S., Hernández, N., Martínez-Pérez, A. M., Nieto-Cid, M., Álvarez-Salgado, X. A., Baños, I., Montero, M. F., Mazuecos, I. P., Gasol, J. M., Osterholz, H., Dittmar, T., Berman-Frank, I. and Arístegui, J.: Basin-wide N2 fixation in the deep waters of the Mediterranean Sea, Glob. Biogeochem. Cycles, 30, 1–19, doi:10.1002/2015GB005326.Received, 2016.*

3. Why would fixed N inputs add to this area only if diazotrophy is related to water masses which are moving around the ocean? Is this really only a locally important processes add N to this area only?

*It is difficult to speculate here, but in principle fixed N2 (into ammonium or DON) would be consumed in a short time by the in situ bacterial community. In a recent opinion paper (now in review in Frontiers in Marine Science), we estimate that $N_2$ fixed and eventually remineralized to nitrate in the mesopelagic zone would turn over in 4 to 43 years. Please see our response to this reviewer's comment on the same issue below.*

4. Unclear why the depth is reported as dbar here. I suggest the authors change dbar to meters.

*It was a general agreement among all the scientists involved in the OUTPACE cruise to use dbar in all of our publications for consistency with CTD files and easy exchange of data among groups.*

Figure 1. I suggest that oxygen is reported as umol L-1 or umol kg-1 and not mL L-1 which is an unconventional unit for oxygen on oceanography. This figures is not clear because it is not possible to see the specific rates of nitrogen fixation here. I suggest this is replotted to show the actual values for nitrogen fixation, which would be more useful considering the uniqueness of this data set.

*In our previous aphotic N2 fixation studies we plotted rates as sized dots with oxygen concentrations (color scale) in the background. Since no significant relationships were found between aphotic N2 fixation and oxygen concentrations during the OUTPACE cruise, we agree with this reviewer that it may be more reasonable to plot rates in a different way. We now provide aphotic N2 fixation rates as sized dots (as we find it very visual and easy to spot where activity is higher), but with actual rates superimposed in coloured numbers (see Figure 2 in the response to reviewers file)..*

[Figure]

Figure 5. This is not clear due to words in blue overlapping as well as SD5 to SD15 overlapping. Can this be replotted, e.g as colour codes?
*We agree that the relative positioning of overlapping samples within the gray box was not clear, however, assigning distinct color or shape codes to the samples would not resolve this problem as some samples are directly over one another. Thus, one of the conclusions from this figure is that these samples are very similar to each other, driven by the relationship to PC1 of the DOM analysis (which they fall under). The variation among samples within the gray box is extremely small when compared to the distance to other samples, which differ based on the various factors shown. We have addressed the issue of overlapping text by removing the text from the grey box, and are now showing the sample names only in the zoom-out box on the left. Other text in the figure was also made smaller, which improved overall readability.*

Figure S1. The DIN and phosphate around station 7 look odd? There is no DIN and phosphate between 400 and 1000m. Please check.
*Indeed, this was an error. DIN, DIP and several variables are considered core parameters shared among all researchers participating in the OUTPACE cruise special issue in Biogeosciences. We have therefore decided to refer the reader to figures 5a-b in Fumenia et al.'s paper (same issue) https://www.biogeosciences-discuss.net/bg-2017-557/bg-2017-557.pdf , where nitrate and phosphate concentrations are shown.*

Table S1 has fallen off the bottom of the page. Please explain in the legend how to interpret the numbers. Are these p values or is a high value good, i.e. means a strong relationship. What do the stars mean?
*We apologize for this. In the current version we have re-dimensioned the table so that it does not fall off the page. One asterisk means significant correlation at the 0.05 level, two asterisks mean significant correlation at the 0.01 level. This information has been provided in the table caption.*

Minor details/comments: Abstract, line 29: remove 'here'. Change of tense, suggest 'we measured....and identified...'
*Corrected as suggested.*

This sentence is awkward 'Because non-cyanobacterial diazotrophs presumably need external dissolved organic matter (DOM) sources for their nutrition, we also identified DOM compounds using Fourier Transform Ion Cyclotron Mass Spectrometry (FTI-CRMS)' - suggest change to 'DOM sources were identified.....because non-cyans...

*We agree that this sentence was rather incomplete. We have rewritten it as follows:*

*"Because non-cyanobacterial diazotrophs presumably need external dissolved organic matter (DOM) sources for their nutrition, we also identified DOM compounds using Fourier Transform Ion Cyclotron Resonance Mass Spectrometry (FTICRMS) with the aim of searching for relationships between the composition of DOM and non-cyanobacterial N2 fixation in the aphotic ocean."*

Page 2, line 1: remove majorly
*We have replaced it with "mostly", in order to conserve the meaning of the sentence.*

Page 2, line 8: '....that aphotic N2 fixation may contribute significantly to fixed nitrogen inputs in this area.' As above....Why just this area? Considering the deep ocean consists of water masses moving water and its properties around the ocean, what would the nitrogen fixation here contribute to the N budget here only?
*It is difficult to speculate here, but in principle fixed N2 (into ammonium or DON) would be consumed in a short time by the in situ bacterial community. In a recent opinion paper (now in review in Frontiers in Marine Science), we estimate that $N_2$ fixed and eventually remineralized to nitrate in the mesopelagic zone would turn over in 4 to 43 years. We have however rephrased it to:*

*"While the data available is still too scarce to elucidate the distribution and controls of mesopelagic non-cyanobacterial diazotrophs in the WTSP, their prevalence in the mesopelagic layer and the consistent detection of active N2 fixation activity at all depths sampled during our study suggest that aphotic N2 fixation may contribute significantly to fixed nitrogen inputs in this area and/or areas downstream of water mass circulation."*

Page 3: Line 17: the N2 fixation rate should be removed as a volumetric rate rather than integrated rate. For example, it may only be high because it is integrated over a thick layer of the ocean?
*With this sentence we intended to highlight the importance of the WTSP as a hotspot of photic N2 fixation worldwide. As noted in the text:*

*"The WTSP has been recently recognized as a global hotspot of photic N2 fixation, harboring among the highest N2 fixation rates ever recorded (~600 μmol N m-2 d-1; Bonnet et al., 2017), mostly attributed to Trichodesmium and UCYN-B (Berthelot et al., 2017; Bonnet et al., 2015; 2009; Stenegren et al., 2017)."*

*These photic measurements correspond to the integration of rates obtained at 5 to 7 levels in the sunlit layer, and therefore we believe it is legitimate to present it as integrated rates.*

Page 5: Line 5: 'measured the initial δ15N of N2 in the incubation on each incubation bottle by membrane inlet mass spectrometry analyses (MIMS; Kana et al., 1994)' - do you mean after the addition of 15N2? Then this needs to be clearer here. But range of enrichments were you achieving here? In light of the newness of this approach, it would be appropriate to include some detail here.
*MIMS samples were taken at the end of incubations. The $^{15}N$ at% values obtained were $7.548 \pm 0.557$ at% (Bonnet et al., 2018). We have rewritten this part of the M&M as follows:*

*"To obtain accurate N2 fixation rates we (1) measured the δ15N of background N2 in the incubation on each incubation bottle by membrane inlet mass spectrometry analyses (MIMS; Kana et al., 1994) -the values obtained were $7.548 \pm 0.557$ at% (Bonnet et al., 2018)-,"*

*Bonnet, S., Caffin, M., Berthelot, H., Grosso, O., Benavides, M., Helias-Nunige, S., Guieu, C., Stenegren, M. and Foster, R. A.: In depth characterization of diazotroph activity across the Western Tropical South Pacific hot spot of N2 fixation, Biogeosciences, (January), 1–30, doi:10.5194/bg-2017-567, 2018.*

Page 11: Note that Tricho colonies have been detected in sediment traps elsewhere, e.g. Pabortsava et al 2017 in Nature Geosciences
*We have cited Pabortsava et al. as well.*

**Reviewer #3**

The study by Benavides and coauthors report aphotic N2 fixation rates and identify diazotrophs present in the mesopelagic layer of the western tropical South Pacific. The paper is a significant contribution which increases the knowledge about aphotic nitrogen fixation in a region which is highly interesting in terms of N-input from N2 fixation. Rates of N2 fixation were low but detected across all depths and stations. Shifts in diazotroph assemblages seemed to be mostly associated with depth. A distinct 1G phylotype was identified to coincide with the oxygenated Sub-Antarctic Mode Water. The paper is very well written and the methods used are well described, solid and established.

*We thank this reviewer for his/her positive comments.*

In my opinion the presentation of data could be improved by clearer links to different water masses. In the title the oligotrophic to ultraoligotrophic transect is highlighted but the way that this translates into sampling stations and different water masses is not evident to the reader from the figures. Further the nifH data is presented largely based on depth rather than sampling location/water mass.

*In this study, our objective was to sample throughout the mesopelagic zone, not necessarily targeting any specific water masses. The depths sampled (200, 500, 650 and 800 db) were "arbitrarily" chosen according to water volume availability in deep casts during the OUTPACE cruise (note that we needed as much as 40 L per depth to perform all our analyses). Very interestingly, when examining the nifH sequencing results, it turned out that a specific phylotype was predominant in a given water mass (subcluster 1G in the SAMW). Unfortunately, the coverage of our samples throughout the mesopelagic zone is not enough to represent all the different water masses present and to identify patterns in N2 fixation activity or diversity of diazotrophs according to water mass distribution.*

*This can be clearly seen in the T-S diagrams shown in Figure 1 (from the response to reviewers file).. On the left, we present a T-S diagram of the water masses sampled during the OUTPACE cruise (as displayed in Fig. 4a in Fumenia et al., this issue). According to this T-S diagram, our N2 fixation and nifH gene measurements (central and right figures) correspond to the lower part of the upper thermocline ($\sigma_t$=24.7-25.4), lower part of the lower thermocline ($\sigma_t$=26.5-26.7), and SAMW/AAIW ($\sigma_t$=26.7-27.3). No measurements are available in the two water masses of the central thermocline.*

[Figure]

The DOM analysis is valid but considering the low abundances these diazotroph groups are likely present in compared to other members of the microbial community establishing connections may be difficult. From the results section is not evident if differences in DOM compounds were seen across the transect or different depths.

*Our group investigated aphotic N2 fixation and its relationship with DOM in a couple cruises in the Solomon Sea (Benavides et al., 2015) and in the Mediterranean Sea (Benavides et al., 2016), in the frame of the* project DIADOM *https://cordis.europa.eu/project/rcn/187917_en.html*
*In both cases we found positive correlations between labile compounds and N2 fixation. In the OUTPACE cruise we basically followed the same sampling strategy, but did not find significant relationships between DOM composition and aphotic N2 fixation. Although the FTICRMS data may itself not add much to the present study,, we decided to keep it for comparison with our previous studies and to reinforce the need for a mechanistic understanding of how non-cyanobacterial diazotrophs interact with DOM in the ocean.*

*Benavides, M., H. Moisander, P., Berthelot, H., Dittmar, T., Grosso, O. and Bonnet, S.: Mesopelagic N2 fixation related to organic matter composition in the Solomon and Bismarck Seas (Southwest Pacific), PLoS One, 10(12), 1–19, doi:10.1371/journal.pone.0143775, 2015.*

*Benavides, M., Bonnet, S., Hernández, N., Martínez-Pérez, A. M., Nieto-Cid, M., Álvarez-Salgado, X. A., Baños, I., Montero, M. F., Mazuecos, I. P., Gasol, J. M., Osterholz, H., Dittmar, T., Berman-Frank, I. and Arístegui, J.: Basin-wide N2 fixation in the deep waters of the Mediterranean Sea, Glob. Biogeochem. Cycles, 30, 1–19, doi:10.1002/2015GB005326.Received, 2016.*

The (relatively) high N2 fixation rates at station 13 are curious and could be given some more attention in the discussion. From Fig 2. it appears like the diazotroph composition from station 13 differs largely between the depths and clusters away from the other samples. I find it very intriguing that this suggests that several different groups may be responsible for similar rates at the different depths. It is mentioned that high concentrations of chlorophyll were observed at this station. Did this coincide with high photic N2 fixation rates?

*Indeed, it is very interesting that aphotic N2 fixation rates were highest at station 13. These high rates coincided with a patch of chlorophyll at the surface (de Verneil et al., 2017; this issue), and the high photic N2 fixation rates (see figure 2e in Bonnet et al. (2018), same issue). We hypothesized that labile organic matter exported from the photic zone fuels aphotic N2 fixation below, as we previously observed in another cruise in the WTSP (Benavides et al., 2015). Within station 13, the 200 and 650 dbar samples clustered closely with other samples from the same depths. It was interesting that at 500 dbar, the majority of the nifH sequences were from cluster 3Q, and at 800 dbar, from 1O. These high proportions of the community are due mostly to specific OTUs (denovo18755 and denovo6047, respectively), which are included in Figure 4.*

*Benavides, M., H. Moisander, P., Berthelot, H., Dittmar, T., Grosso, O. and Bonnet, S.: Mesopelagic N2 fixation related to organic matter composition in the Solomon and Bismarck Seas (Southwest Pacific), PLoS One, 10(12), 1–19, doi:10.1371/journal.pone.0143775, 2015.*

*Bonnet, S., Caffin, M., Berthelot, H., Grosso, O., Benavides, M., Helias-Nunige, S., Guieu, C.,*

*Stenegren, M. and Foster, R. A.: In depth characterization of diazotroph activity across the Western Tropical South Pacific hot spot of N2 fixation, Biogeosciences, (January), 1–30, doi:10.5194/bg-2017-567, 2018.*

*de Verneil, A., Rousselet, L., Doglioli, A. M., Petrenko, A. A. and Moutin, T.: The fate of a southwest Pacific bloom: Gauging the impact of submesoscale vs. mesoscale circulation on biological gradients in the subtropics, Biogeosciences, 14(14), 3471–3486, doi:10.5194/bg-14-3471-2017, 2017.*

Other comments: The presentation of average N2 fixation rates and relation to % of photic N2 fixation is unclear and values in abstract and text appear to be different. (Abstract Lines 33-34 and Discussion Lines 20-23)

*We agree. The overall contribution of aphotic N2 fixation (across the whole transect) ranges between 6 and 88% (as said on the abstract). In the discussion we split the contribution to the two regions (Melanesian Archipelago -MA-, and subtropical gyre -GY-), with average contributions of 13 and 51%, respectively (according to the regional values provided in Bonnet et al., 2018).*

*Bonnet, S., Caffin, M., Berthelot, H., Grosso, O., Benavides, M., Helias-Nunige, S., Guieu, C., Stenegren, M. and Foster, R. A.: In depth characterization of diazotroph activity across the Western Tropical South Pacific hot spot of N2 fixation, Biogeosciences, (January), 1–30, doi:10.5194/bg-2017-567, 2018.*

The Bray-Curtis distances in Figure 2 might be more meaningful if done on a level with higher resolution. Currently the variations in phylotypes is largely "hidden" in the 1G subcluster. A rarefaction to equal sampling depth would further improve this analysis.

*The clustering was conducted using the method by Frank et al. (2016), which uses the subclustering to the lowest level of 1G. This method thus does not allow going to a higher resolution with subcluster 1G. By resampling to only 15,000 reads (average 89,000, std dev 42,500), there is no significant difference in Shannon or Simpson diversity indices. Additionally, we have previously compared similar sequence data analyzed via resampling to the same sequencing depth and the results were not appreciably different. Additional resolution of distributional patterns within Cluster 1G is shown in the OTU based analysis shown in Figure 2.*

*Frank, I. E., Turk-Kubo, K. A. and Zehr, J. P.: Rapid annotation of nifH gene sequences using classification and regression trees facilitates environmental functional gene analysis, Environ. Microbiol. Rep., 8, 905–916, doi:10.1111/1758-2229.12455, 2016.*

In Figs. 2 and 3 data is presented as depth but in Fig. 1 as pressure [dbar]

*It should be dbar. Figs. 2, 3 and 4 have been corrected accordingly.*

Fig. 1 Please adjust the scale so that the circles are not cut for stations 1 and 15

*In order to address the comments of Reviewer #2, we have changed Fig. 1 providing aphotic N2 fixation rates as sized dots (as we find it very visual and easy to spot where activity is higher), but with actual rates superimposed in coloured numbers (see Figure 2 from the response to reviewers file).*